# Electrostatics in semiconducting devices I : The Pure Electrostatics Self Consistent Approximation

A. Lacerda-Santos* and Xavier Waintal

*Univ. Grenoble Alpes, CEA, IRIG-PHELIQS GT, F-38000 Grenoble, France*†

In quantum nanoelectronics devices, the electrostatic energy is the largest energy scale at play and, to a large extend, it determines the charge distribution inside the devices. Here, we introduce the Pure Electrostatic Self consistent Approximation (PESCA) that provides a minimum model that describes how to include a semiconductor in an electrostatic calculation to properly account for both screening and partial depletion due to e.g. field effect. We show how PESCA may be used to reconstruct the charge distribution from the measurement of pinch-off phase diagrams in the gate voltages space. PESCA can also be extended to account for magnetic field and calculate the edge reconstruction in the quantum Hall regime. The validity of PESCA is controlled by a small parameter $\kappa = C_g/C_q$, the ratio of the geometrical capacitance to the quantum capacitance, which is, in many common situations, of the order of 1%, making PESCA a quantitative technique for the calculation of the charge distribution inside devices.

This article is the first of a series of three articles [1, 2] addressing the electrostatic modeling of quantum nano-electronic devices. It introduces certain concepts that will be extended in the second article of the series in order to provide a robust path for solving the self-consistent quantum-electrostatic problem (SCQE, also known as self-consistent Poisson-Schödinger). The last article of the series deal with PESCADO, an open source code that implement these algorithms.

## I. INTRODUCTION

Quantum transport problems are often studied within simple models where the electric potential seen by the conducting electrons is defined as a given external potential instead of being calculated. This approach can provide interesting insights on certain physical mechanisms. However, it suffers from two important limitations. First, certain effects require to take into account the screening effect of the conducting electrons even at a qualitative level. A famous example is the reconstruction of the edge states in the quantum Hall effect into alternated compressible and incompressible stripes [3–6]. Second, and perhaps more importantly, it prevents the modeling from being able to predict quantitatively important features as simple as the conductance versus gate voltage characteristics of a device. With the advent of new generations of increasingly complex devices for emerging quantum technologies (e.g. spin qubits or Majorana qubits), being able to make simulations with true predictive power is rapidly becoming a necessity. There has been important recent efforts towards this goal [7–21].

SCQE is a mean field approach that can provide this type of calculation. The problem consists of three parts

---

* antoniosnl@hotmail.com
† xavier.waintal@cea.fr

which we formally write as the Schrödinger equation,

$$[H_0 - eU]\Psi_{\alpha E} = E\Psi_{\alpha E} \qquad (1)$$

where $\psi_{\alpha E}(\vec{r})$ is the electronic wave function at energy $E$ and the discrete index $\alpha$ labels the different sub-bands (or propagating channels) that propagate in the device. $H_0$ is the Hamiltonian and $U(\vec{r})$ the electric potential. From the wave function, one obtains the electron density $n(\vec{r})$,

$$n(\vec{r}, \mu_\alpha) = \int \frac{1}{2\pi} |\Psi_{\alpha E}(\vec{r})|^2 f_\alpha(E) \qquad (2)$$

where $f_\alpha(E) = 1/[e^{(E-\mu_\alpha)/kT_\alpha} + 1]$ is the Fermi function in the corresponding electrode with electro-chemical potential $\mu_\alpha$ and temperature $T_\alpha$. At zero temperature and equilibrium ($\mu_\alpha = \mu$ and $T_\alpha = 0$), Eq.(2)) is essentially the Integrated local density of states (ILDOS) and we shall abusively refers to it as the ILDOS even in the general case. Finally, the Poisson equation closes the set of equation,

$$\nabla \cdot (\epsilon(\vec{r})\nabla U(\vec{r})) = en(\vec{r}) - en_{\mathrm{d}}(\vec{r}), \qquad (3)$$

where $e > 0$ is the electron charge, $\epsilon(\vec{r})$ the local dielectric constant and the doping charge density $n_{\mathrm{d}}$ accounts for electric charges not treated at the quantum mechanical level, e.g. dopants of surface charges.

Despite its apparent simplicity, the set of equations (1),(2) and (3) is not always easy to solve numerically. The combination of long range interaction with intrinsic non-linearities often prevent simple iterative schemes from converging. The most popular techniques so solve this problem iterate through the sequence Eq.(1) $\rightarrow$ (2) $\rightarrow$ (3) $\rightarrow$ (1) $\rightarrow$ ... and use various kinds of predictors/correctors to force the convergence [22–44]. These approaches work well when the ILDOS $n(\vec{r}, \mu_\alpha)$ grows roughly linearly with $\mu$ (such as at high temperature) but often fail to converge in presence of strong non-linearities. An alternative route has been put forward in [7] where a local non-linear problem is solved exactly

and the treatment of the long range part is improved iteratively. The approach has been very successful in strongly non-linear situation [6]. Here we develop yet another point of view that aims at treating the long range and non-linear aspects simultaneously. Our approach is based on a parametrization of the ILDOS function (as a function of $\mu$) in terms of a piece wise linear function where all the non-linearities are concentrated in the kinks between different parts. For such a parametrization, the self-consistency can be done exactly in a finite number of iterations. Once the approximate problem is solved, one does not iterate on the density or electric potential as is usually done but directly on the ILDOS function to reach the solution of the SCQE problem.

In this first paper, we take the first step in this direction and show how one can solve the self-consistent problem where the ILDOS is approximated by a piecewise linear function. This naturally leads us to introduce an approximate SCQE problem that we call the Pure Electrostatic Self Consistent Approximation (PESCA). We show that PESCA captures the salient features of interest in actual devices within a accuracy better than $\sim 1\%$ in many situations.

The rest of this article is organized as follows. First we introduce the small parameter $\kappa$ of the SCQE problem - it estimates the error made when approximating the density in the 2DEG as controlled only by the electrostatics of the device. Then in Section III we formulate the PESCA and propose an algorithm to solve the SCQE problem under PESCA for any 1D, 2D or 3D models. In Section IV we apply PESCA to study a concrete example, the split quantum wire system shown in Fig.1. We calculate the associated "pinch-off phase diagram" for the quantum wire system, i.e. the various thresholds for which parts of the 2DEG gets depleted, see Fig.5. We show how to use the calculated phase diagram to calibrate the microscopic parameters of our model. In Section V we look into the inverse problem of Section IV, i.e. we suppose we are given an experimental pinch off diagram and propose a method to extract the microscopic parameters for our model. In Section VI we extend the PESCA algorithm to solve the self-consistent problem for the Integer Quantum Hall effect ILDOS. We recover the compressible/incompressible stripes configuration first proposed by Chklovskii Shklovskii and Glazman [3–5] and compare our calculations for the QHE to those we made using the algorithm we have proposed in an earlier publication [6].

## II. THE SMALL PARAMETER OF THE SCQE PROBLEM

Let's consider a SCQE problem simple enough to be solved in a few lines of algebra. We consider an infinite two-dimensional electron gas (2DEG), such as the one found in GaAs/GaAlAs heterostructures. The 2DEG is situated at a distance $d$ from the surface, where we place a metallic electrode. A voltage $V_g$ is applied be-

tween the metallic electrode and the 2DEG with a voltage source. Setting the zero of the energy in the metallic gate, the gate voltage sets the electro-chemical potential in the 2DEG $\mu = eV_g$. In the effective mass approximation, the density of states in the 2DEG is a constant $\rho = sm^*/(2\pi\hbar^2)$ where $s$ is the generacy of the gas ($s = 2$ for spin, $s = 4$ for spin and valley degneracy). The solution of the quantum part of the problem reduces to a simple spatially independent ILDOS,

$$n = \rho(eV_g - eU) \tag{4}$$

where the electric potential $U$ in the the 2DEG sets the position of the bottom of the conduction band. The electrostatic problem reduces to a simple planar capacitor,

$$n = \frac{\epsilon}{ed}U \tag{5}$$

Introducing the geometrical capacitance (per unit surface) $C_g = \epsilon/d$ and the quantum capacitance $C_q = e^2\rho$ one arrives at,

$$n = \frac{1}{1+\kappa}\frac{C_g}{e}V_g \quad \text{with} \quad \kappa = \frac{C_g}{C_q} \tag{6}$$

For most semiconducting systems, the dimensionless parameter $\kappa$ is actually very small $\kappa \ll 1$. Therefore from Eq.(6) we conclude that to good approximation the density in the 2DEG is given by $n \approx C_gV_g/e$, i.e the density in the 2DEG is entirely controlled by the electrostatics.

Table I shows an estimate of $\kappa$ in a few common situations (GaAs and Silicon devices). We find that $\kappa \approx 1-2\%$ is the common situation. The only significant deviation comes from 2DEG that are extremely close to the gates such as in Silicon MOSFET where only a thin oxide seperates the two. This means that the pure electrostatic approximation, $\kappa = 0$ limit, makes an error of only $\approx 1 - 2\%$ in many common situations. The PESCA that we will develop in this article is essentially the generalization of the $\kappa = 0$ limit to a spatially dependent problem. Like the $\kappa = 0$ limit, the PESCA result is independent of the details of the quantum problem, e.g. the effective mass of the material, and as such clarifies how these (geometrical and quantum) parameters affect the physics.

## III. THE PURE ELECTROSTATIC SELF-CONSISTENT APPROXIMATION: PESCA

Before we introduce the PESCA, let's define a concrete electrostatic problem that will be used for illustrations. We consider an infinite split quantum wire defined in a GaAs/GaAlAs 2DEG. A side view of the system is shown in Fig.1. The device is directly inspired by the experiments [10, 45, 46] on split wires aiming at demonstrating flying qubits. Using the top side electrodes, one can deplete the underneath gas and define a wire. Using the central electrode, one can further split this wire into two, potentially leaving some tunneling coupling between the two sub-wires. Besides the 2DEG, the model

| Semi. | $\frac{m^*}{m_e}$ | $\frac{\epsilon}{\epsilon_0}$ | Deg. | $d$ [nm] | $C_q$ [mF/m²] | $C_g$ [mF/m²] | $\kappa$ |
|---|---|---|---|---|---|---|---|
| GaAs (e) | 0.067 | 12.9 | 2 | 100. | 45 | 1.1 | 2.6% |
| Si (e) | 0.2 | 11.7 | 4 | 20 | 268 | 5 | 1.9% |
| Si (e) | 0.2 | 11.7 | 4 | 5 | 268 | 21 | 7.7% |
| Si (h) | 0.49 | 11.7 | 2 | 20 | 327 | 5 | 1.5% |

TABLE I. Typical values of the $\kappa$ parameter for a few common 2DEG. Deg. is the degeneracy (2 for spin, 4 for spin and valleys in the silicon conduction band). e stands for electron gas and h for holes.

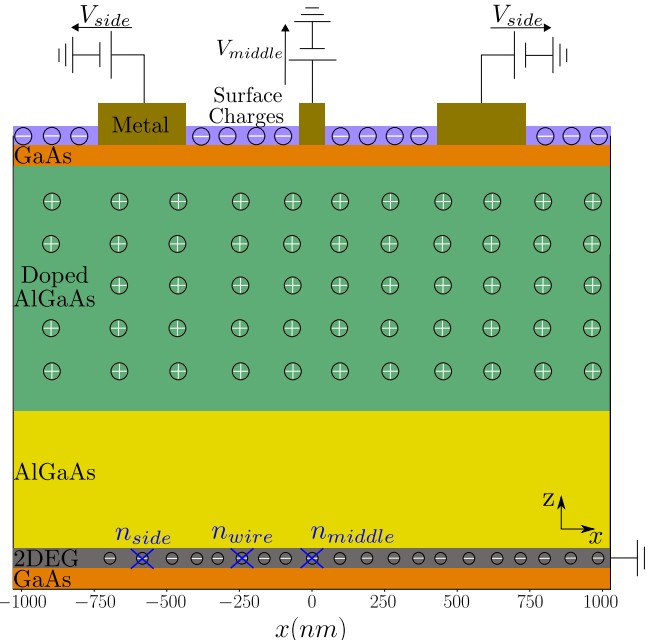

FIG. 1. Side view of the split quantum wire system studied in this article. Various colors correspond to different regions. The system is infinite along the $y$ axis. The orange and yellow regions correspond to $GaAs$ and $AlGaAs$ respectively. The green region corresponds to the doped $AlGaAs$ layer. A finite density of surface charges appears at the GaAs/vacuum upper interface and is shown in violet. Metallic electrodes are shown in brown and correspond to equipotential. The electron density at special points $x_{\mathrm{side}} = -585$nm, $x_{\mathrm{wire}} = -225$nm and $x_{\mathrm{middle}} = 0$nm are called $n_{\mathrm{side}}$, $n_{\mathrm{wire}}$ and $n_{\mathrm{middle}}$ respectively. We use different scales along the $x$ and $z$ directions (See text).

contains positive dopants of concentration $n_{\mathrm{dop}}$ $[m^{-3}]$ in the green region and surface charges of concentration $n_{\mathrm{sc}}$ $[m^{-2}]$ at the free surface of the stack. The metallic gate correspond to electric equipotentials at voltage $V_{\mathrm{side}}$ and $V_{\mathrm{middle}}$ respectively. The dimensions of the different layers are, from top to bottom : 7.5 nm (GaAs cap layer), 88 nm (doped AlGaAs) and 49.3 nm (AlGaAs).

### A. Formulation of PESCA

Fig.2a) shows an example of the ILDOS calculated by solving Eq.(1) and (2) explicitly for a finite width 2DEG in the effective mass approximation. The curve is ap-

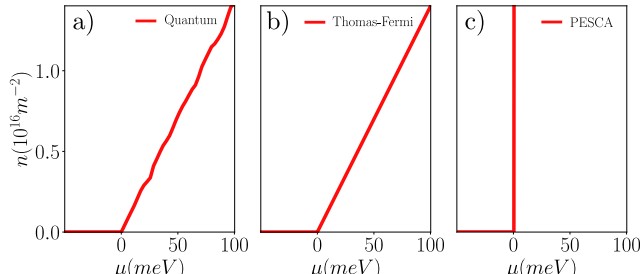

FIG. 2. Example of ILDOS. a): ILDOS obtained from a quantum calculation. b) ILDOS corresponding to Thomas-Fermi for a 2DEG without magnetic field. c) ILDOS in the PESCA approximation. The latter is independent of the material.

proximately linear above the bottom of the band with occasional small $\sqrt{\mu}$ kinks due to the opening of a conducting channel. It also has small variations between different positions $x$ in the 2DEG. It should be noted however, that the biggest source of non-linearity is the large kink at the bottom of the 2DEG conduction band $\mu = 0$. Fig.2b) shows the ILDOS in the Thomas-Fermi approximation. Instead of solving the quantum problem for a finite system, one uses the bulk density of states calculated for a translation invariant 2DEG. Since the density of states of the infinite 2DEG is constant, the Thomas-Fermi ILDOS is linear for $\mu > 0$, with a slope equal to the averaged one of Fig.2a).

The PESCA approximation amounts to taking the slope of the ILDOS to be infinite, i.e. the ILDOS is made of a horizontal branch $n(\mu < 0) = 0$ and a vertical branch $\mu(n > 0) = 0$, as shown in Fig.2c). For a bulk system it is equivalent to setting the small parameter $\kappa = 0$. Keeping only the vertical branch $\mu = 0$ would correspond to a good metal with a density of states high enough for the conductor to always remain an equipotential. Keeping only the horizontal branch would correspond to a fully depleted 2DEG. By keeping both branches, one allows for partially depleted devices, i.e. for a proper treatment of field-effect transistors. In particular, PESCA can predict the "pinch-off" phase diagrams, i.e. the values of the gate voltages where the gas is depleted underneath the side gates or the middle gates in Fig.1. Solving the PESCA problem amounts to finding which part of the 2DEG is on the horizontal/vertical branch and solve the

associated electrostatic problem.

As argued in the preceding section, the PESCA is expected to be quantitative (within a few percent) for the calculation of thermodynamic quantities such as the density. Its level of validity for more subtle observables such as the conductance remains to be asserted. The relevance of PESCA lies in several features. (i) First, as we shall see, PESCA can be calculated very easily, at a much smaller computational cost than the full treatment of the SCQE problem. (ii) PESCA can help solving SCQE problems more accurately by treating different parts of the system at different level of description. For instance, suppose one wants to solve the SCQE problem of our split wire, c.f Fig.1, in a regime where the 2DEG is depleted below the side gates so that the wire is formed. One could treat the 2DEG inside the wire region $|x| < |x_{\text{side}}|$ by solving the corresponding quantum problem. However, it is inefficient to treat the 2DEG outside of the wire region in the same way, as it would be very costly computationally. On the other hand, it is necessary to take into account that part of the 2DEG as it partially screens the effect of the gates. Treating this outside region $|x| > |x_{\text{side}}|$ within PESCA would provide a precise and efficient compromise with an automatic calculation of the position of the boundary of the wire. (iii) PESCA can quantitatively (i.e. within a few percent) predict the voltages needed to deplete the 2DEG in different regions (e.g. underneath the side or middle gates). With these depletion voltages we construct what we call "Pinch-off" phase diagrams, e.g. Fig.5. Pinch-off phase diagrams can be easily measured experimentally. They depend on the geometry of the device as well as on the charge distribution in the stack. As such, PESCA calculations may be very valuable to reconstruct this distribution and setup precise electrostatic models. Finding what is the correct distribution of e.g. ionized donors or surface charges, is indeed a prerequisite for predictive simulations of these type of devices [8]. (iv) PESCA forms the basis onto which more precise algorithms for treating Thomas-Fermi problems or SCQE problems will be built [1].

### B. PESCA algorithm

To solve the PESCA problem, one needs to extend traditional Poisson equation solvers such that they allow one to dynamically update the status of the 2DEG so that some parts of it belong to the horizontal branch (depleted) while the rest belongs to the vertical branch of the ILDOS (not depleted).

Following Ref.[7] we first discretize the Poisson equation into a finite volume discrete problem where the system is divided in to small cells $i$ of finite volume. Other discretization schemes such as finite differences or finite elements could be use as well. One arrives at,

$$\sum_j C_{ij}U_j = Q_i. \tag{7}$$

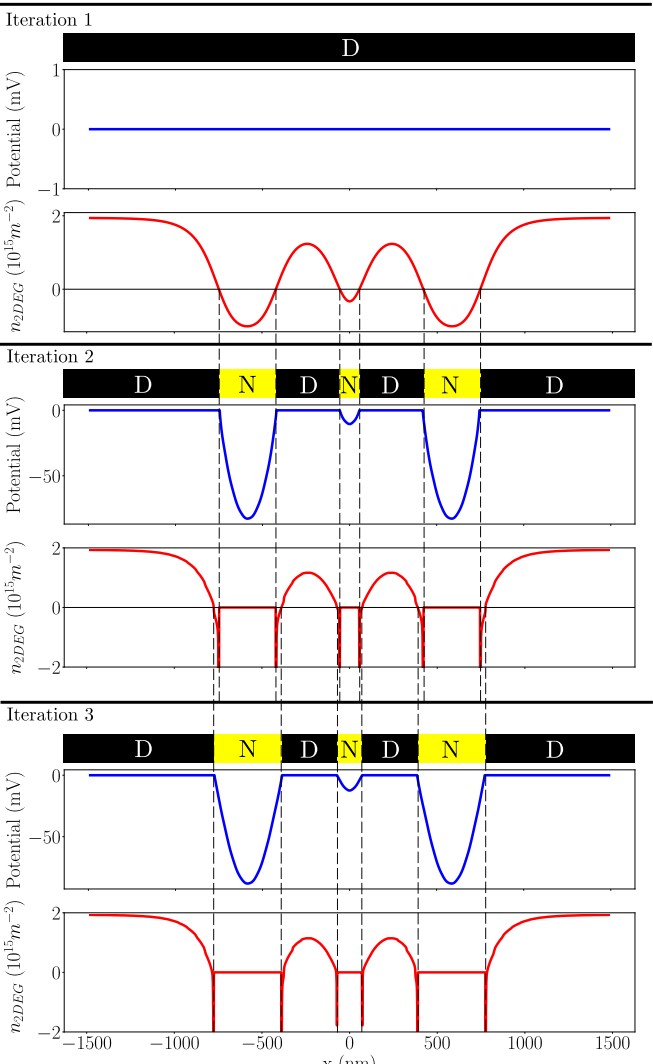

FIG. 3. Illustration of the PESCA algorithm. For each iteration, the upper panel shows the $\mathcal{D}/\mathcal{N}$ partitioning (black for $\mathcal{D}$ cells, $\mathcal{N}$ for yellow cells), the middle panel shows the potential $U(x)$ for this partitioning (blue curve) and the lower panel shows the corresponding density profile $n(x)$ (red curve). The results shown here correspond to a full Dirichlet initialization. We set $V_{mid} = -0.88V$ and $V_{gate} = -0.52V$.

where $C_{ij}$ is a discretized version of the Laplacian operator, $U_i$ the electric potential at the center of cell $i$ and $Q_i$ the number of electrons inside the cell :

$$Q_i = -\int_{C_i} d\vec{r}[n(\vec{r}) - n_{\text{d}}(\vec{r})] \tag{8}$$

Cells that belong to the 2DEG can be sorted out in two categories. For those belonging to the horizontal part of the ILDOS, we know the density ($n_i = 0$) and want to calculate the potential $U_i$. We call these cells the Neumann cells ($\mathcal{N}$). The cells on the vertical branch of the ILDOS are of a different type that we call Dirichlet ($\mathcal{D}$). In these cells we know the electric potential $U_i = 0$ and want to calculate the density $n_i$. Writing Eq.(7) in

a block form for the Dirichlet ($\mathcal{D}$) and Neumann ($\mathcal{N}$) blocks, it reads

$$\begin{bmatrix} C_{\mathcal{NN}} & C_{\mathcal{ND}} \\ C_{\mathcal{DN}} & C_{\mathcal{DD}} \end{bmatrix} \cdot \begin{bmatrix} U_{\mathcal{N}} \\ U_{\mathcal{D}} \end{bmatrix} = \begin{bmatrix} Q_{\mathcal{N}} \\ Q_{\mathcal{D}} \end{bmatrix}. \tag{9}$$

Rewriting the above equation so that the known inputs are on the right hand side and the unknowns on the left hand side, we get a new linear problem that can be solved using standard routines for solving linear problem with sparse matrices,

$$\begin{bmatrix} C_{\mathcal{NN}} & 0 \\ C_{\mathcal{DN}} & -1 \end{bmatrix} \cdot \begin{bmatrix} U_{\mathcal{N}} \\ Q_{\mathcal{D}} \end{bmatrix} = \begin{bmatrix} 1 & -C_{\mathcal{ND}} \\ 0 & -C_{\mathcal{DD}} \end{bmatrix} \cdot \begin{bmatrix} Q_{\mathcal{N}} \\ U_{\mathcal{D}} \end{bmatrix}. \tag{10}$$

The algorithm for solving PESCA solves Eq.(10) iteratively as follows. (I) First, we partition the 2DEG between $\mathcal{N}$ and $\mathcal{D}$ cells. For instance, one may set all the cells to be $\mathcal{D}$, which assumes the 2DEG is nowhere depleted. (II) Second, we solve Eq.(10) to calculate $U_{\mathcal{N}}$ and $Q_{\mathcal{D}}$. (III) Third, we determine the new partitioning of the cells. To do so we proceed as follows. For the $\mathcal{D}$ cells,

- if $Q_i > 0$ then the cell remains a $\mathcal{D}$ cell.

- if $Q_i \leq 0$ then the cell is in the wrong branch of the ILDOS and one must change it to $\mathcal{N}$.

For the $\mathcal{N}$ cells,

- if $U_i < 0$ then the cell remains a $\mathcal{N}$ cell.

- if $U_i \geq 0$ then the cell is in the wrong branch of the ILDOS and one must change it to $\mathcal{D}$.

We repeat steps II and III until the partitioning is stable. Since there is a finite number of partitioning, the algorithm is guaranteed to converge in a finite number of iterations. In practice the convergence is extremely fast.

Fig.3 shows the first three iterations of the above algorithm for the split wire geometry. Starting from a full Dirichlet initialization in iteration-1, we calculate the associated density (red curve) and find that it is negative below the gates that have been polarized with a negative voltage. In iteration-2, we first update the $\mathcal{D}/\mathcal{N}$ partitioning and assume that cells with $Q(x) < 0$ are actually depleted. Then we calculate a new density and potential profile. Since some cells are now depleted, the density close to them is slightly affected (those cells used to screen the metallic gates and do not do so anymore) and the position of the $\mathcal{D}/\mathcal{N}$ interface moves slightly in iteration-3. After a few iterations, the partitioning converges to its final form. The full set of iterations is shown in Fig.4, where the fraction of Dirichlet cells is shown as a function of the iteration number for two different initializations. At 5 iterations most cells are converged and both $U(x)$ and $Q(x)$ profiles are calculated with good precision. Within 12 iterations the cell partitioning is fully converged. We find that for a random initialization with equal fraction of Dirichlet and Neumann cells

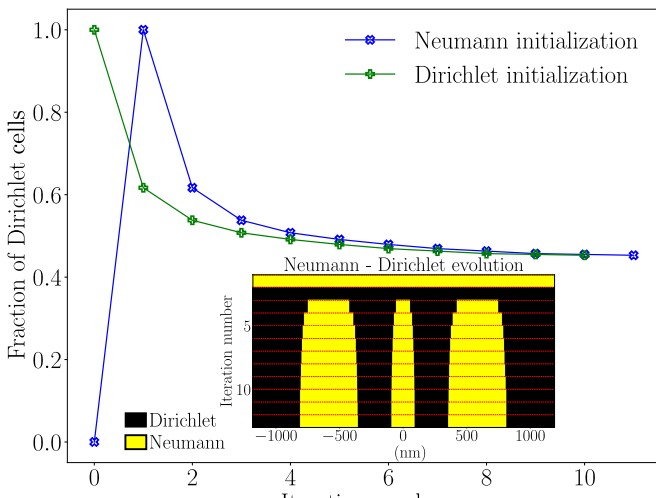

FIG. 4. fraction of Dirichlet cells ($\mathcal{D}$) in the 2DEG as a function of the number of PESCA iterations. In green all cells belonged to $\mathcal{D}$ at the first iteration. In blue all cells were set to Neumann ($\mathcal{N}$) in the initial configuration. The inset on the right corner shows the $\mathcal{D}/\mathcal{N}$ partitioning as a function of position $x$. The black regions correspond to Dirichlet cells and yellow to Neuman points.

convergence is achieved within the same number of iterations. The inset of Fig.4 shows the evolution of the partitioning with the number of iterations. We identify in yellow the depleted regions and in black the non depleted regions.

## IV. APPLICATION TO PINCH-OFF PHASE DIAGRAMS

We now turn to a practical application of PESCA for the split wire geometry. The electrostatic model of such a wire contains several parameters and many of them are not well known experimentally. For instance the density of dopants is only approximately known during the epitaxial growth, and even if known, the fraction of the dopants that are actually ionized is difficult to assess. Additionally, a large fraction of the electrons coming from the dopants actually go to surface states. Hence the actual fraction of ionized donor charge populating the conducting region of the device is difficult to estimate. Another parameter is the workfunction of the metal used in the gates, as it should give rise to offsets on the gate voltages.

These unknown parameters determine the charge distribution in the sample. We argue that, to a large extent, these parameters can be inferred from a combination of measurements and PESCA calculations. Such a program was already performed in the context of the quantum point contact geometry in [8]. Here we concentrate on how the PESCA calculation can be made and fitted to the experimental findings

To study the split wire device with PESCA we use a 2 stages model, a high temperature model and a low temperature model. At high temperature, the surface of the sample is modeled by an equipotential at a voltage $V_{sc}$ and the metallic gate by an equipotential at a voltage $V_{off}$. On the other hand we describe the dopant region by its concentration of dopants $n_{dop}$. If we were to use an equipotential $V_{dop}$, it would amount to supposing that the phenomena of "Fermi level pinning" happens in the dopant region. Fermi level pinning can happen when the concentration of dopants is high enough so that charges can move from one dopant to another (at high temperature) and screen the electrostatic gates, see the discussion in [8]. Such an effect is not considered here. We first solve the high temperature Poisson problem and calculate the distribution of charge $n_{sc}(x)$ at the surface. At low temperature, we freeze the surface charge. That is, we turn

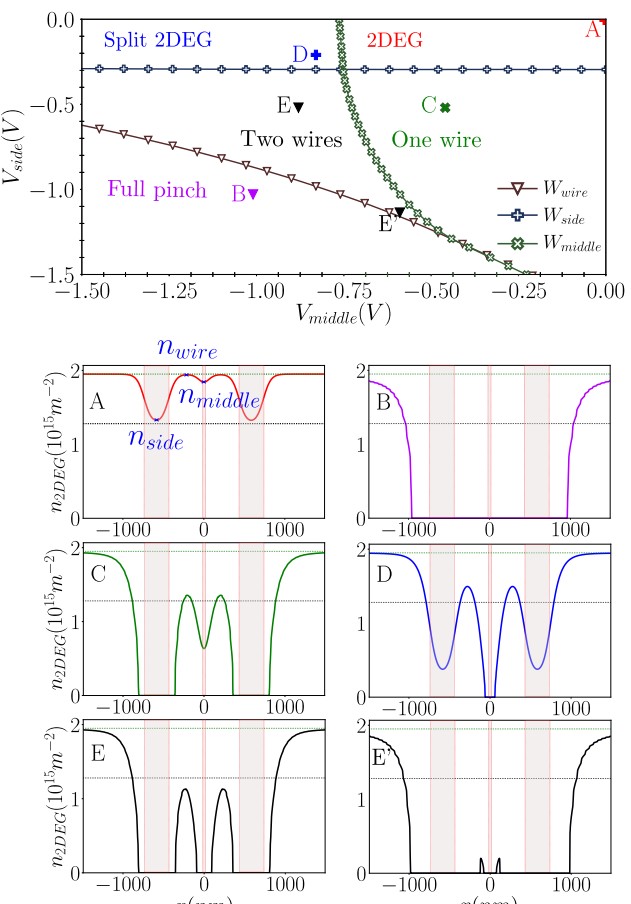

FIG. 5. Top panel: PESCA Pinch-off phase diagram for the split wire device shown in Fig.1. $n_{dop} = 1.4310^{16}m^{-2}$, $n_s = 1.2810^{15}m^{-2}$, $V_{off} = -0.813V$ and $V_{sc} = -0.668V$. The different lines $W_{wire}$, $W_{side}$ and $W_{middle}$ separate the different regions A-E, see text. Bottom panels: 2DEG density profile calculated for the points A-E in the phase diagram. The labels $n_{side}$, $n_{middle}$ and $n_{wire}$ indicate the 2DEG density at specific points, respectively, underneath the side gate, the middle gate and in between, see Fig.1.

the surface from $\mathcal{D}$ to $\mathcal{N}$ with a distribution of charges given by $n_{sc}(x)$. Among the regions outside of the 2DEG, only the gates remain $\mathcal{D}$ at the low temperature model. We note $n_s$ the bulk 2DEG density far away from the gates. Although $n_s$ is calculated, it is easily measured experimentally with e.g. Hall resistance. One can adjust the dopant density to obtain the correct value of $n_s$.

Fig.5 shows the PESCA "Pinch-off" phase diagram of the model for a particular set of parameters $n_{dop}$, $V_{sc}$ and $V_{off}$. For each region, a density profile is shown in the bottom panels for a particular $(V_{side}, V_{middle})$ point. In region A, the 2DEG is present underneath all gates. Note however that even though no gate voltage is applied in point A, the density underneath the gates is reduced due to the workfunctions of the electrodes. In region B, the 2DEG is entirely depleted in the central part of the split wire.

When the side gate voltage is negative enough, the 2DEG underneath the side gates get depleted and the quantum wire is formed (region C). Upon further applying a negative voltage on the middle gate, one eventually cuts the wire into two, thus reaching the split wire regime (region E). In region D, only the gas underneath the central gate is depleted. At the same time, the different lines of the phase diagram depend on the model parameters. Hence, pinch-off phase diagrams can put strong constraints on the underlying modeling. All the while pinch-off phase diagrams can also be easily measured experimentally. For example, they can be constructed by measuring the conductance between ohmic contacts separated by the different regions of the split wire device (e.g. conductance of each subwire or between the two subwires). Then, the different separation lines in the diagram correspond to the gate voltages for which the corresponding conductance reaches zero.

Fig.6 shows the density at $x_{wire} = -225nm$ (middle of the left subwire) as a function of the gate voltage (the same voltage is applied to all the gates). First, these curves are not linear. This is a consequence of the self-consistent nature of PESCA. Indeed for a given, fixed $\mathcal{D}/\mathcal{N}$ partitioning, the density at every point depends linearly on the gate voltage. Hence the non-linearity is a consequence of the variation of this partitioning as some regions get depleted. The different curves of Fig.6 (black, green and blue) are calculated by varying the gate voltage applied at the high temperature stage of the calculation. This is aimed at mimicking a common experimental practice known as "bias cooling", where a positive bias is applied during the cooling of the sample. This bias attract electrons under the gate which in turn are screened by charges elsewhere in the device. In our model, it is the surface charges that screen the electrons under the gate. At low temperature these screening charges get frozen, hence remain even if the bias voltage is removed. They contribute to depleting the 2DEG underneath the gate even at zero voltage. We find that this effect is qualitatively reproduced by our calculations. However, typically a one to one correspondence is found experimentally be-

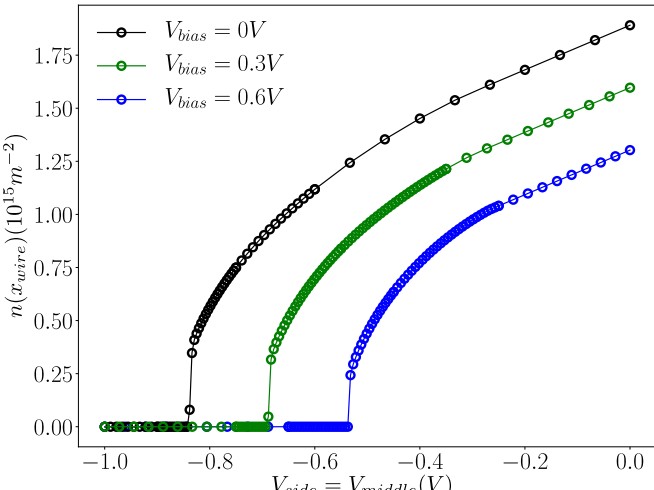

FIG. 6. Effect of bias cooling. Density $n(x_{\text{wire}l})$ versus gate $V_{\text{side}} = V_{\text{middle}}$ at $x_{\text{wire}} = -225nm$ as a function of $V_{bias}$. The $n(V)$ profiles are all calculated at the low temperature stage but for different $V_{bias}$ at the high temperature stage. The different curves correspond respectively to black for $V_{bias} = 0V$, green for $V_{bias} = 0.3V$ and blue for $V_{bias} = 0.6V$.

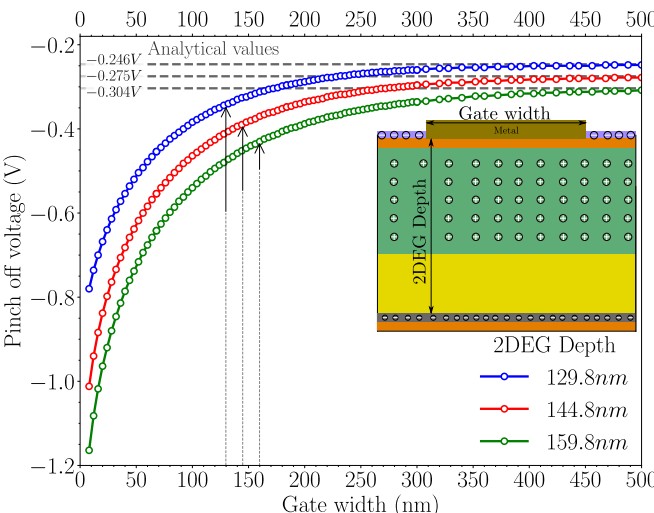

FIG. 7. Pinch off voltage in a single gate geometry (see inset) versus gate width. The parameters are $n_{dop} = 1.43 \; 10^{16}m^{-2}$, $V_{off} = -0.813V$ and $V_{sc} = -0.668V$. The different curves correspond to three different distances between the gate and the 2DEG obtained by increasing the thickness of the undoped $AlGaAs$ layer

tween the bias used during cooldown and the gate value at which pinch-off is found at low temperature [47]. In our calculations the offset in the pinch-off value is only that of half the cooldown bias voltage. This points to the presence of additional screening effects in the donor layer not taken into account here. For instance, we have supposed that the donor ionization is fully frozen at high temperature.

Fig.7 shows the pinch off voltage in a simpler geometry

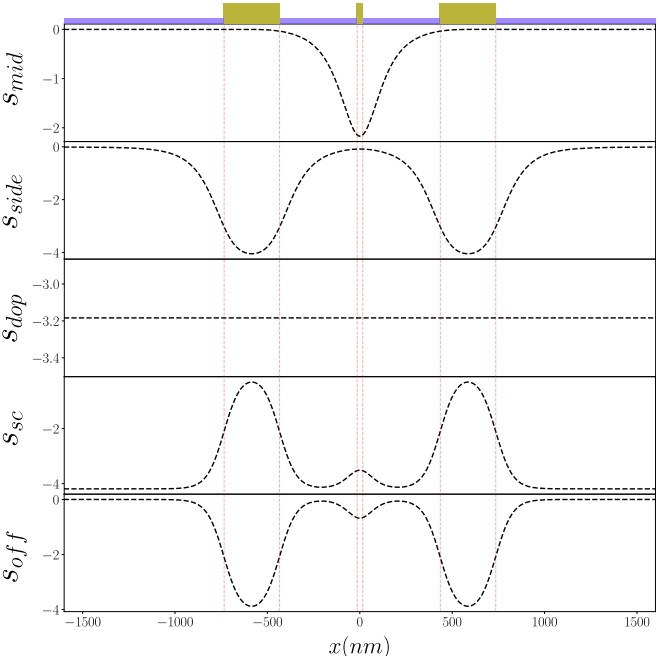

FIG. 8. Functions $\mathbf{s}_{\text{mid}}(\mathbf{x})$, $\mathbf{s}_{\text{side}}(\mathbf{x})$, $\mathbf{s}_{\text{off}}(\mathbf{x})$, $\mathbf{s}_{\text{sc}}(\mathbf{x})$ providing the full solution of PESCA in region A of the phase diagram, see Eq.(11). The vertical red dotted lines show the positions of the electrostatic gates.

where only a single gate is present. When the gate width is larger than roughly 200-300 nm (about twice the distance from the 2DEG to the gate), one approaches the bulk value of the pinch-off (which can trivially be calculated analytically). However, the Pinch-off value changes quite strongly when the gate is made thinner. Since in actual devices this width may be as thin as 30 nm or perhaps even thinner, we find that the geometry must be accounted for precisely if one wants to make quantitative predictions.

## V. ADJUSTING THE PINCH-OFF PHASE DIAGRAM TO EXPERIMENTAL DATA

In this section we look at the inverse problem to the one from the preceding section: suppose that we are given an experimental Pinch-off phase diagram, how do we extract the microscopic parameters $n_{\text{dop}}$, $V_{\text{sc}}$ and $V_{\text{off}}$ of the model ? Of course, one could design some optimization problem and minimize the difference between the target pinch-off phase diagram and the one calculated from PESCA. However such a step is rather computationally intensive and lacks physical insights. Besides, further insight can be obtained by using the particularly simple structure of PESCA in some limiting regimes, as we show below.

The first limit of interest corresponds to the region A of the phase diagram of Fig.5. In this region, where the gate voltages are only weakly negative, the 2DEG is depleted

| Regions | PESCA | | Region A Full metallic limit | | Region B Depleted limit | | | |
| --- | --- | --- | --- | --- | --- | --- | --- | --- |
| | | | s_mid s_side | s_sc s_dop s_off | s_sc s_dop s_off | U_mid U_side | U_sc U_dop U_off | |
| | High T | Low T | Low T | High & Low T | High T | Low T | High T | Low T |
| Surface charge | $\mathcal{D}$ | $\mathcal{N}$ | $\mathcal{N}$ | $\mathcal{D}$ | $\mathcal{D}$ | $\mathcal{N}$ | $\mathcal{D}$ | $\mathcal{N}$ |
| Middle & Side gates | $\mathcal{D}$ | $\mathcal{D}$ | $\mathcal{D}$ | $\mathcal{D}$ | $\mathcal{D}$ | $\mathcal{D}$ | $\mathcal{D}$ | $\mathcal{D}$ |
| Dopants | $\mathcal{N}$ | $\mathcal{N}$ | $\mathcal{N}$ | $\mathcal{N}$ | $\mathcal{N}$ | $\mathcal{N}$ | $\mathcal{N}$ | $\mathcal{N}$ |
| 2DEG | $\mathcal{D}/\mathcal{N}$ | $\mathcal{D}/\mathcal{N}$ | $\mathcal{D}$ | $\mathcal{D}$ | $\mathcal{D}$ | $\mathcal{N}^*$ | $\mathcal{D}/\mathcal{N}$ | $\mathcal{D}/\mathcal{N}$ |

TABLE II. Summary of the boundary conditions we fix at each region of the model throughout this article. $\mathcal{D}$ means the cells are of Dirichlet type. $\mathcal{N}$ means the cells are of Neumann type. $\mathcal{D}/\mathcal{N}$ means the cells can be of either Dirichlet or Neumann type.
* The 2DEG far from the side gates are of $\mathcal{D}$ type instead of $\mathcal{N}$ (See text)

nowhere. Hence all the cells are $\mathcal{D}$ cells and as such no self-consistency is required. This is the fully metallic limit. In the Fully metallic limit, the 2DEG density is the solution of a single linear system of the form of Eq.(10). From the additivity of different solutions, it results that $n(x)$ is a linear function of the different input parameters,

$$
\begin{aligned}
n(x) = & V_{\text{middle}} s_{\text{mid}}(x) + V_{\text{side}} s_{\text{side}}(x) \\
& + V_{\text{off}} s_{\text{off}}(x) + V_{\text{sc}} s_{\text{sc}}(x) + \frac{n_{\text{dop}}}{10^{16}} s_{\text{dop}}(x)
\end{aligned}
\quad (11)
$$

where $s_{\text{mid}}(x)$, $s_{\text{side}}(x)$, $s_{\text{off}}(x)$, $s_{\text{sc}}(x)$ and $s_{\text{dop}}(x)$ are functions to be determined. Each function can be obtained from a single call to the linear solver. In other words we can get the full parametric dependance of $n(x)$ with respect to the input parameters. An example of the $s_{\text{mid}}(x)$ set of functions is given in Fig.8. The frontiers of the Pinch-off phase diagram that connect region A to another region can be obtained directly from Eq.(11). The frontier $W_{\text{middle}}$ between region A and D is given from $n(x = x_{\text{middle}} \equiv 0) = 0$, which translates into

$$
\begin{aligned}
& V_{\text{middle}} s_{\text{mid}}(0) + V_{\text{side}} s_{\text{side}}(0) \\
& + V_{\text{off}} s_{\text{off}}(0) + V_{\text{sc}} s_{\text{sc}}(0) + \frac{n_{\text{dop}}}{10^{16}} s_{\text{dop}}(0) = 0
\end{aligned}
\quad (12)
$$

Similarly, the frontier $W_{\text{side}}$ between region A and region C is given from $n(x = x_{\text{side}} \equiv -585nm) = 0$ in Eq.(11). If these frontiers are known experimentally, we directly obtain a set of linear constraints on the microscopic parameters that facilitate the fitting of the model to the experiments.

The calculation of the set of functions $s(x)$ require a little care with respect to the boundary conditions. For instance the workfunction $V_{\text{off}}$ can naively be thought as a voltage which could be simply added to $V_{\text{side}}$ and $V_{\text{middle}}$. However, $V_{\text{off}}$ is present at high temperature where the surface charges are mobile (i.e. modeled by $\mathcal{D}$ cells) while the gate voltages $V_{\text{side}}$ and $V_{\text{middle}}$ are set at low temperature when the surface charges are frozen (i.e.

modeled by $\mathcal{N}$ cells). Table II summarizes the boundary conditions we fix at each calculation we performed in this article. For instance, it shows that to calculate $s_{\text{mid}}$ we set the surface charge region to $\mathcal{N}$, the middle & side gates region to $\mathcal{D}$, the dopants region to $\mathcal{N}$ and the 2DEG region to $\mathcal{D}$.

Another interesting limit is the "depleted limit". It corresponds to region B of the Pinch-off diagram. In this regime there is no electron gas inside the central split wire region. However, some electron gas remains in the region $|x| > |x_{\text{dep}}|$ far away from the central region. In PESCA, the position $x_{\text{dep}}$ is determined self-consistently. However, it is interesting to consider the case where $x_{\text{dep}}$ is fixed, as it allows us to study the screening role of the electron gas far away from the central region. For a fixed value of $x_{\text{dep}}$, the potential inside the wire is given by

$$
\begin{aligned}
U(x) = & V_{\text{middle}} U_{\text{mid}}(x) + V_{\text{side}} U_{\text{side}}(x) \\
& + V_{\text{off}} U_{\text{off}}(x) + V_{\text{sc}} U_{\text{sc}}(x) + \frac{n_{\text{dop}}}{10^{16}} U_{\text{dop}}(x)
\end{aligned}
\quad (13)
$$

where $U_{\text{mid}}(x)$, $U_{\text{side}}(x)$, $U_{\text{off}}(x)$, $U_{\text{sc}}(x)$ and $U_{\text{dop}}(x)$ are functions to be determined. To determine the frontier between region B and the other regions from Eq.(13), one can use

$$
\exists\, x, \text{ such that } |x| < |x_{\text{dep}}| \text{ and } U(x) = 0. \quad (14)
$$

In contrast to region A, where the position of depletion is expected to be directly under the gates, in region B the position where the gas will appear depends on the ratio $V_{\text{side}}/V_{\text{middle}}$. Therefore the corresponding $W_{\text{wire}}$ curve is not a simple straight line. Much like for $s(x)$, calculating the set of $U(x)$ functions require special care with respect to the boundary conditions, as summarized in Table II. This is specially true at the 2DEG. At low temperature the 2DEG is made of $\mathcal{N}$ cells while at high temperature it is made of $\mathcal{D}$ cells. Therefore, calculating, e.g. $U_{\text{sc}}(s)$, requires solving *two* different linear systems: one to determine $s_{\text{sc}}$ and a second to calculate the effect of $s_{\text{sc}}$ when the 2DEG is depleted.

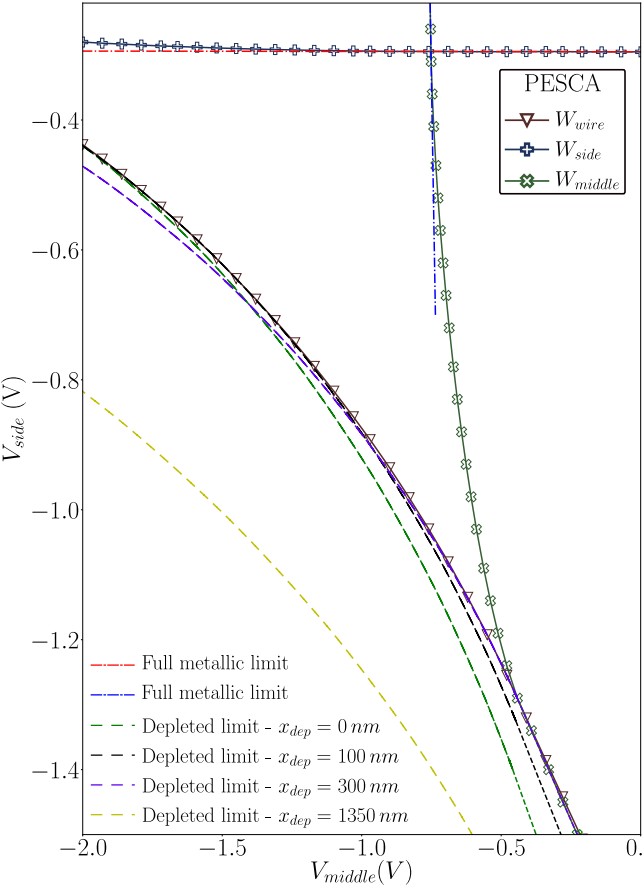

FIG. 9. Comparison between PESCA Pinch-off phase diagram with the full metallic limit and the depleted limit. The full lines correspond to PESCA results obtained with $n_{dop} = 1.43 \, 10^{16} m^{-2}$, $n_g = 1.95 \, 10^{15} m^{-2}$, $n_s = 1.28 \, 10^{15} m^{-2}$, $V_{off} = -0.813V$ and $V_{sc} = -0.668V$. The red and blue dotted lines correspond to calculations in the full metallic limit. The green, black, violet and yellow dotted lines correspond to calculations in the depleted limit as one varies $x_{dep}$. Here $x_{dep}$ delimits the distance between the last depleted 2DEG cell and the side gates.

Fig.9 shows the PESCA phase diagram together with the predictions obtained from the full metallic limit and the depleted limit. The prediction from the full metallic limit is essentially exact. Its appeal stems from the fact that (in contrast to a PESCA calculation) we have the full dependance of $n(x)$ on $n_{dop}$, $V_{sc}$ and $V_{off}$. The prediction from the depleted limit depends strongly on the value of $x_{dep}$. This means the occupied 2DEG outside of the split wire region plays an important screening role, which cannot be ignored if one wants to make quantitative predictions.

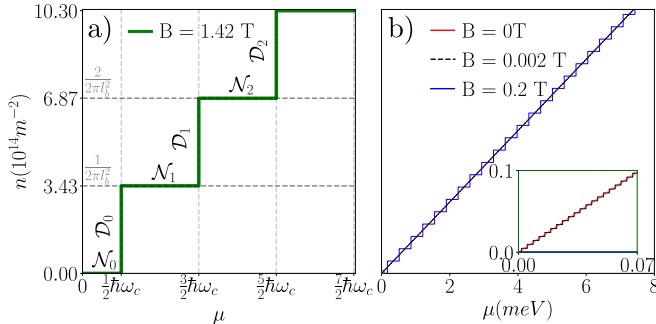

FIG. 10. a) Thomas Fermi ILDOS for $B = 1.4T$. One can see the $\mu$ and $n$ regions for which a site is of $\mathcal{N}_i$ or $\mathcal{D}_i$ type up to $i = 2$. For example, if $1/2\hbar\omega_c < \mu < 3/2\hbar\omega_c$ then the site is of $\mathcal{N}_1$ type with $n = 1.81.10^{14} m^{-2}$. b) Thomas Fermi ILDOS for $B = 0.2T$ and $B = 2mT$ in blue and black respectively. The ILDOS in red corresponds to Thomas-Fermi at $B = 0T$. At the energy scales relevant to the problem in question there is no discernable difference between the $B = 2mT$ and the Thomas-Fermi ILDOS

## VI. EXTENSION OF PESCA TO THE INTEGER QUANTUM HALL EFFECT

We end this article with the first extension of the PESCA algorithm to more complex situations. As we have seen, PESCA essentially relies on the ILDOS having two different branches. However, it is straightforward to generalize the algorithm so that it can handle an ILDOS with a discrete number of branches. Such a situation arises naturally in the quantum Hall regime when the 2DEG is put under a perpendicular magnetic field. The role of electron-electron interactions is known to have drastic effects in this regime and in particular leads to the reconstruction of the edge states into compressible and incompressible stripes [3–5]. The density of states of a bulk 2DEG in the quantum Hall regime is indeed highly non-linear. It is made of delta function peaks in the (highly degenerate) Landau levels separated by insulating regions of vanishing density of states. The resulting ILDOS is a step like function as shown in Fig.10a. In Fig.10a, $\hbar\omega_c = 2.45$ meV, with $\omega_c$ the cyclotron frequency, while $l_B = 21.6$ nm is the magnetic length. To solve the self consistent problem corresponding to this ILDOS, one updates the PESCA algorithm by introducing the cells $\mathcal{D}_p$ and $\mathcal{N}_p$. The integer $p \geq 0$ indicates on which branch of the ILDOS the cell is. The $\mathcal{D}_p$ cells are Dirichlet cells with fixed potential $U = \hbar\omega_c(p + 1/2)$ while the $\mathcal{N}_p$ cells are Neumann cells with fixed density $n = p/(2\pi l_B^2)$. Step III of the PESCA algorithm also needs to be extended as follows. For the $\mathcal{D}_p$ cells,

- if $n_i > (p+1)/(2\pi l_B^2)$ then the cell must change to $\mathcal{N}_{p+1}$

- if $n_i < (p/(2\pi l_B^2))$ then the cell must change to $\mathcal{N}_p$

- otherwise the cell remains in $\mathcal{D}_p$.

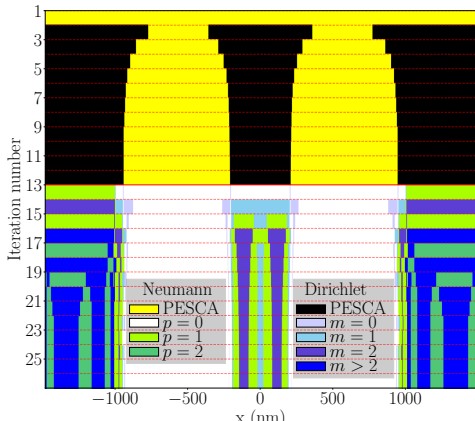

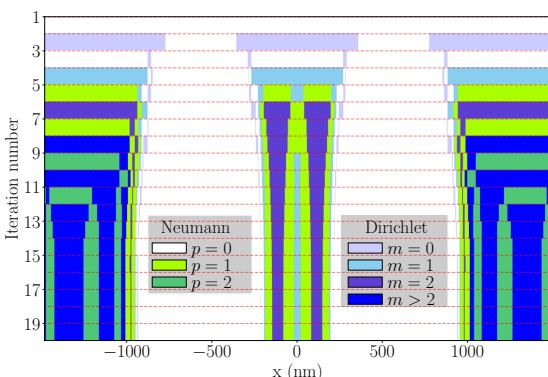

FIG. 11. $\mathcal{N}_p/\mathcal{D}_p$ partitioning as a function of the iteration number for the results in Fig.12. (a) Up to iteration 12 the PESCA approximation was used. At the latter iteration the PESCA $\mathcal{N}/\mathcal{D}$ partitioning is stable. The resulting potential and charge profile are then used as input to iteration 13. For the latter and onwards thomas-fermi approximation is used for $B = 1.42T$. The yellow and black regions correspond to $\mathcal{N}$ and $\mathcal{D}$ cells under PESCA approximation. Under thomas fermi the cells located within the first five landau levels are shown in varying degrees of green for $\mathcal{N}_p$ cells and blue for $\mathcal{D}_p$ cells. $p$ is the filling factor s.t. $n = p/(2\pi l_b)$ and $m$ is defined s.t. $p = (1 + m)/2\hbar\omega_c$. (b) The thomas fermi approximation for $B = 1.42T$ is used for all iterations.

For the $\mathcal{N}_p$ cells,

- if $U_i \geq -\hbar\omega_c(p + 1/2)$ then the cell must change it to $\mathcal{D}_{p-1}$.

- if $U_i \leq -\hbar\omega_c(p + 3/2)$ then the cell must change it to $\mathcal{D}_p$.

- otherwise the cell remains in $\mathcal{N}_p$.

Fig.11 shows the convergence of the the $\mathcal{N}_p$ / $\mathcal{D}_p$ partition for the extended PESCA. On the top Fig.11 an initial zero magnetic field PESCA (two-branch ILDOS Fig.2 (c)) calculation is performed before switching on the magnetic field and performing a second calculation

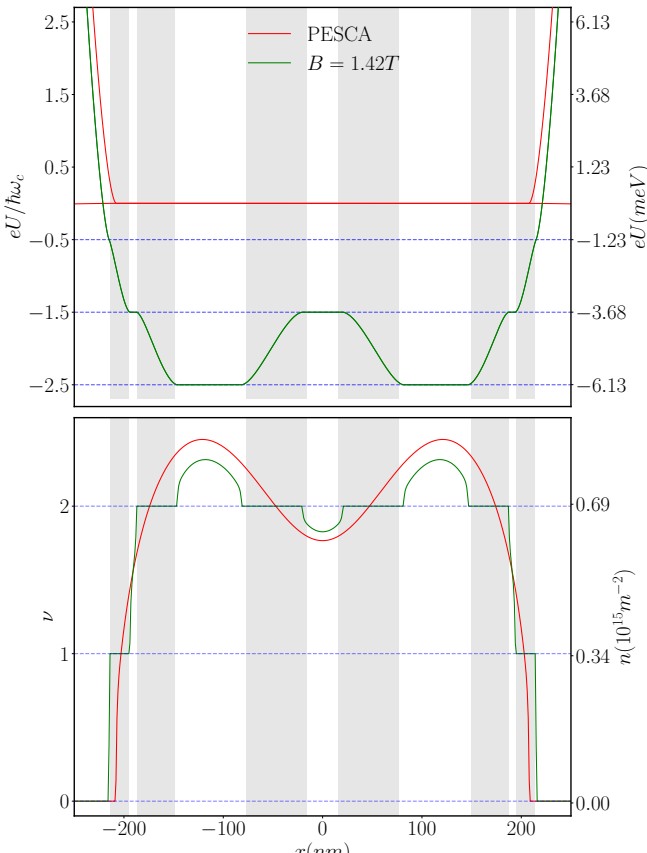

FIG. 12. Top panel : Chemical potential profile at the 2DEG as a function of $x$. The profile in green was obtained for $B = 1.4T$ with the Ildos on Fig.10 (a). In red the profile was obtained with the PESCA approximation, i.e. the Ildos on Fig.2 (c). The gray (white) stripes correspond to the incompressible (compressible) region. Lower panel : Charge density profile for $B = 1.4T$ and with the PESCA approximation. At the regions where the charge density is depleted, the red curve converges towards the green one. For this calculation $V_{side} = -1V$ and $V_{mid} = -0.35V$.

with extended PESCA. On the bottom Fig.11 the iterations are directly done with a finite magnetic field. We find that both schemes converge although starting with a simple PESCA initialization seems to speed up the convergence. This is particularly the case at very low magnetic fields where many Landau levels are filled. The converged partition gives direct access to the so-called compressible ($\mathcal{D}_p$) and incompressible ($\mathcal{N}_p$) stripes [3–5].

Fig.12 shows the converged profile of electric field (top) and density (bottom). We refer to [6, 7] for a discussion of the physics of the different stripes present in the system. We note that, even though the magnetic field is fairly high B=1.42T (cyclotron energy $\hbar\omega_c = 2.45meV$), the density profile is only weakly affected by the field with respect to PESCA ($\approx 5\%$). Also, the modification of the electric potential of a few mV is small compared to the larger scales at play within the rest of the sample, of the order of 1V. These few mV might be associated

Fermi calculation corresponding to the ILDOS of Fig.2b that was performed using the algorithm of [7]. The direct Thomas-Fermi calculation is indistinguishable from the $B = 0.002T$ one. This can also be seen in Fig.10b, where the different ILDOS are plotted. Besides being an independent validation of the calculation (the two algorithms only share the Poisson solver), this points to the fact that *any* ILDOS could be approximated by step like functions with $\mathcal{N}_p$ / $\mathcal{D}_p$ partitions, which would allow the PESCA algorithm to be extended to arbitrary Thomas-Fermi approximations (i.e. with fixed ILDOS).

## VII. CONCLUSION

In this paper we have demonstrated two main points regarding electrostatic calculations. First, we took advantage of the existence of a small parameter in the self-consistent quantum electrostatic problem, the ratio of the geometric to quantum capacitance, to develop the Pure Electrostatic Self Consistent Approximation. PESCA is an appropriate level of calculation for reconstructing the charge distribution in a sample, which in turn is a prerequisite for predictive simulations of quantum nanoelectronics devices. PESCA can be thought of as the minimum level at which the 2DEG must be included to account for its screening effect in an electrostatic calculation.

Second, the PESCA algorithm is guaranteed to converge and forms the first step of a new kind of solver for SCQE problems. We have shown that PESCA could be extended to address ILDOS that are made up of many small horizontal and vertical branches. A next step, to be shown in a subsequent publication, is to extend the $\mathcal{N}_p$, $\mathcal{D}_p$ branches to allow for arbitrary slopes of the branches and address any piece-wise linear ILDOS. Once this step is achieved, we shall return to the initial SCQE problems and show that the extensions of the PESCA algorithm provide a fast and robust route to solve them.

### ACKNOWLEDGMENTS

X.W. acknowledges funding the European Union H2020 research and innovation program under grant agreement No. 862683, "UltraFastNano", the ANR DADDI, the ANR TKONDO as well as from the Agence Nationale de la Recherche under the France 2030 programme, reference ANR-22-PETQ-0012.

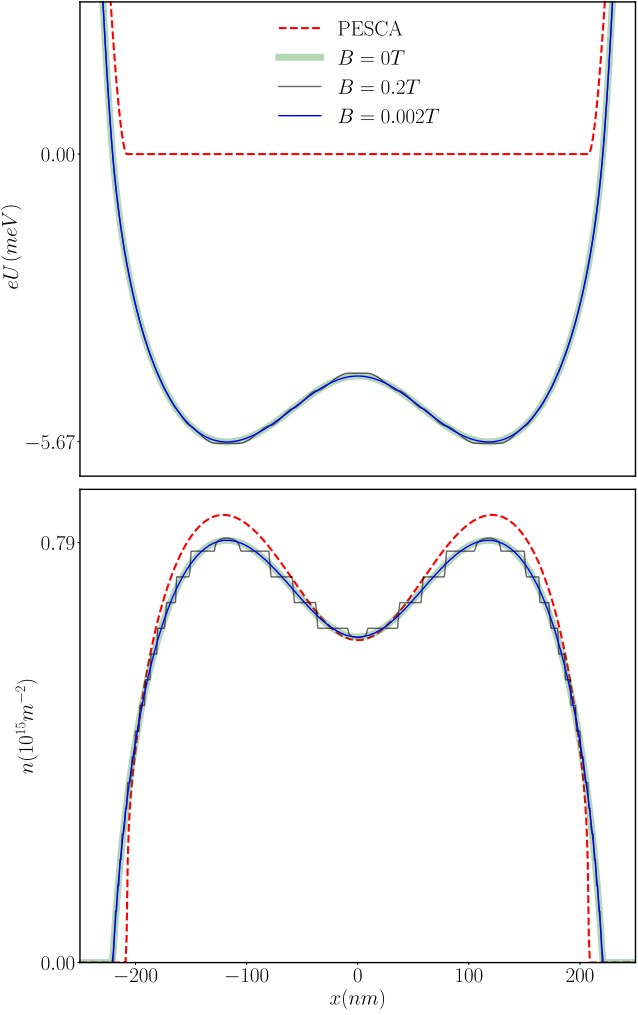

FIG. 13. Top panel : Chemical potential profile at the 2DEG as a function of $x$. In blue and grey the profile was obtained for with $B = 2mT$ and $B = 0.2T$ respectively . They were calculated using the ILDOS on the right panel of Fig.10. In red the profile was obtained with the PESCA approximation, i.e. the Ildos on Fig.2 (c). The thicker green line corresponds to a Thomas Fermi calculation for $B = 0T$ using the algorithm of [7]. Lower panel : Charge density profile

to important physics, but on the other hand they only weakly affect the Pinch-off gate voltages. This confirms that PESCA is an appropriate level of approximation for reconstructing the charge distribution inside the sample.

This can be further seen in Fig.13 where a similar calculation is performed at low $B = 0.2T$ and very low $B = 0.002T$ magnetic field. We also add a direct Thomas

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
