# Peer review of "Electrostatics in semiconducting devices I : The Pure Electrostatics Self Consistent Approximation"

_SciPost Physics_

## Round 3 · Referee Report · Anonymous (Referee 1) · 2025-8-7

Report

The Authors wrote a very interesting article - the first of a promised series of three - on an approximate numerical method for solving the Schrodinger-Poisson problem in semiconducting devices. The method divides the semiconductor in cells which can be either fully depleted or not, is guaranteed to converge, and is rather intuitive to understand. Beyond broadcasting the method with a few applications to a split-wire gated device, also in the presence of a magnetic field, the Authors identify the small parameter that can be used to refine the solution, but postpone the discussion to the next article of the series.

I think this is remarkable work and it probably warrants publication in SciPost Physics. Indeed, I mostly agree with the authors' self-assessment on how this article fulfills journal expectations: this algorithm can alleviate a bottleneck in the simulation of quantum semiconducting devices and it can certainly lead to multi-pronged follow-up work. I have some residual doubts that this is the case, which can probably be resolved by a revision or by some further clarification. Namely:

1) It is hard to judge the impact of PESCA without reading the forthcoming part II of the series, dealing with the self-consistent solution of the problem. The Authors identify a small parameter but it is not clear how a robust iterative method can be based on it starting from PESCA as a zero-order solution. I would welcome a bit of anticipatory details in this first part of the series already.

2) About the small parameter: discussing Figure 2c, the authors write that "For a bulk system it is equivalent to setting the small parameter κ = 0.". I feel this is the key idea of the work and it should be explained in more detail. Why are the two things equivalent? Is the slope in Fig. 2b proportional to 1/κ? But then how should one reconcile with Eq. (6) which for κ = 0 gives a linear relation between n and Vg? Or should one have another intuition altogether behind this step in reasoning?

3) The showcase applications of Sections 3,4,5 reflect other ongoing projects in the group so they seem not to have been chosen in virtue of their simplicity or illustative power. The split-wire geometry is used for a rather advanced application of semiconducting devices (flying electron qubits) and it is not one of the most used device layouts. Furthermore, the demonstration of the algorithm are all purely 1d (with one direction of the 2DEG assumed to be translationally invariant). It would have been very valuable to showcase some 2D examples on commonly used experimental geometries such as a quantum point contact or a laterally confined quantum dot. Perhaps this can be done in future work, but it would at least be good to add a discussion on these potential applications (if they are feasible, or not).

4) In the quantum Hall bar simulations, tackling edge reconstruction, the Authors defer all the physics discussion to previous literature, but I find it hard to understand what is going on in Figure 11 and what is the significance of the results. A more detailed discussion would be welcome here.

5) There are many minor typos and typesetting inconsistenices throughout the paper (e.g. Pinch-off with a capital letter or without, Thomas-Fermi non capitalized, et cetera). These do not impede understanding but they make reading harder, and should be fixed.

Recommendation

Ask for minor revision

---

## Round 3 · Referee Report · Anonymous (Referee 2) · 2025-9-8

Report

The paper is about an approximation to the self-consistent electrostatics problem following the assumption that the geometric capacitance is much higher than the quantum capacitance, or that the density is entirely controlled by the electrostatics. The approximation amounts to using a singular integrated local density of which is vanishing below the band edge and has infinite slope at the band edge. This means that every part of the system is either at potential zero (at the band edge) with varying density, or depleted with varying potential. The authors refer to the the two different branches of the integrated local density of states as corresponding to Dirichlet and Neumann boundary conditions. The PESCA algorithm involves iteratively partitioning the system into Dirichlet or Neumann regions and solving the simplified problem therein.

The PESCA approximation seems useful especially for a first pass. The big missing piece here is the lack of benchmarking. It would be useful to see how the PESCA solution compares to the full self-consistent solution for some example cases. The authors claim that the deviation is small when the parameter Cg/Cq is small, but they do not show this.

Further questions: 1. Why are there only a finite number of partitionings in the PESCA algorithm? 2. Section 5: What are the s and U functions? How are they determined? I could not really follow this section. Perhaps this should be better explained for readers without specialised knowledge.

Recommendation

Ask for minor revision

---

## Editorial Decision

awaiting_resubmission